Phylogeography of Swertia perennis in Europe based on cpDNA markers

Urbaniak Jacek jacek.urbaniak@upwr.edu.pl 1
Kwiatkowski Paweł 2
Pawlikowski Paweł 3
1 Department of Botany and Plant Ecology, Wrocław University of Environmental and Life Sciences , Wrocław , Poland
2 Department of Botany and Nature Protection, University of Silesia in Katowice , Katowice , Poland
3 Department of Plant Ecology and Environmental Conservation, Faculty of Biology, Biological and Chemical Research Centre, University of Warsaw , Warsaw , Poland
Cowling Richard
Electronic publication date: 2018 Sep 17
Publication date: 2018
Volume: 6
Electronic Location ID: e5512
Received 2018 Mar 6; Accepted 2018 Aug 3
Copyright: ©2018 Urbaniak et al.
Copyright year: 2018
Copyright holder: Urbaniak et al.
License: This is an open access article distributed under the terms of the Creative Commons Attribution License, which permits unrestricted use, distribution, reproduction and adaptation in any medium and for any purpose provided that it is properly attributed. For attribution, the original author(s), title, publication source (PeerJ) and either DOI or URL of the article must be cited.
License URL: https://creativecommons.org/licenses/by/4.0/

Keywords: Swertia, Swertia perennis, Carpathians, Alps, cpDNA, Refugia, Haplotypes, Disjunction

Funding: The authors received no funding for this work.

==============================
Background

Swertia perennis(Gentianaceae) is a perennial diploid and clonal plant species that is discontinuously distributed in peat bogs in the mountains of Europe, Asia and North America as well as in the lowlands of Europe. The current geographical dispersion of S. perennis is probably the result of quaternary climatic changes that have played an important role in determining the distribution of Swertia and other plant and animal species.

Methods

In this study we used molecular techniques and combined data from chloroplast DNA markers (trnLF region and trnH-psbA spacer) to elucidate the phylogeography of S. perennis in Europe. Plants were collected from 28 populations in different locations in the lowlands and mountainous areas of Europe (e.g., the Carpathians, Sudetes, Bohemian Forest and Alps). cDNA was analysed to detect the genetic relationship between specimens from different locations.

Results

A total of 20 haplotypes were identified across the dataset. They were characterised by a high level of genetic variability but showed a lack of phylogeographical structure. This pattern may be the result of repeated recolonization and expansion from several areas. Such genetic differentiation may also be attributed to the relatively long-term isolation of S. perennis in Pleistocene refugia in Europe, which resulted in independent separation of different cpDNA phylogenetic lineages and variation in the nucleotide composition of cpDNA.

Discussion

The lack of strong phylogeographical structure makes it impossible to indicate the centre of haplotype diversity; however, refugia located in the Carpathians, Sudetes or Alps are the most probable sites where S. perennis existed in Europe. This lack of structure may also indicate a high level of gene flow in times when the landscape and fen systems were not fragmented in numerous geographically-isolated populations. This makes it difficult to speculate about the relationships between Asiatic and European plant populations and the origin and distribution of this species in Europe. Today, it seems to be restricted due to the occurrence of plants which clearly reflects the genetic variability from the ancient period.

Introduction

The distribution of organisms and their genetic structure are the consequences of repeated Quaternary climatic changes in ecosystems. These changes have dramatically modified the vegetation and have resulted in extinctions in colder areas of Europe, America and the Arctic (Hewitt, 1996; Hewitt, 2004; Taberlet et al., 1998). Climate change was also the primary reason that numerous plants and animals migrated to the southern parts of Europe or warmer localities (before ice cover), where they survived unfavourable conditions before later beginning their remigration to the north (Ronikier, Cieślak & Korbecka, 2008; Slovák et al., 2012). Some of the most commonly-recognised southern refugia are located in the Mediterranean, on the Balkan Peninsula and in isolated mountain ranges (e.g., the Carpathians and Alps). However, the separation of Central and Northern Europe or Northern Russia by a broad belt of lowlands could have harboured numerous plants that inhabited similar sites or ecosystems in the Pleistocene (Schönswetter, Popp & Brochmann, 2006; Paun et al., 2008; Urbaniak, Kwiatkowski & Ronikier, 2018). The Alps were covered by ice during this period (Mojski, 1993), which restricted the distribution of plants to isolated non-glaciated sites (nunataks or plants that were migrating to the Alps as secondary migrants) that were located close to the area’s peripheral refugia (Stehlik, 2000; Stehlik, 2003; Schönswetter et al., 2005; Ronikier et al., 2008). Brockmann-Jerosch & Brockmann-Jerosch (1926) suggested that it was possible for species to survive in scattered localities in the glaciated Alps (i.e., the nunataks at the top of the Alps); however, it is also possible that species migrated into the Alps from more southern refugia or from Eastern Europe (Schönswetter, Popp & Brochmann, 2006). This scenario was confirmed for Ranunculus pygmaeus, which probably migrated to the valleys of the Alps from source populations in Siberia via the Carpathians (Schönswetter, Popp & Brochmann, 2006). Such migrations were possible due to alternating warm and glacial periods in the Quaternary, which allowed gene exchange and plant migration between the European mountain ranges (e.g., the Alps, Carpathians and Sudetes) (Pawłowski, 1928; Ronikier, Cieślak & Korbecka, 2008). In contrast to the Alps, the Carpathians and Sudetes were glaciated only locally or partially during the Quaternary (e.g., the lower massifs generally remained ice-free). The existence of numerous valleys and several ranges more than 2,000 m above sea level meant that they offered a wide spectrum of sites or ecosystems as potential refugia for organisms in the fragmented subranges in several regions in Europe. Outside of the Alps, the Sudetes, the Bohemian Forest and Carpathians also seemed to play an important role as botanical crossroads for the plants that were migrating from Siberia, the Arctic or the Caucasus (Schönswetter, Popp & Brochmann, 2006; Ronikier, Cieślak & Korbecka, 2008).

A phylogeographical hypothesis for analysing the historical climatic processes that influenced the differentiation in population genetics has been investigated intensively in the Alps and Carpathians, thereby linking together studies in the European mountain ranges (e.g., the Alps, Carpathians, Sudetes and the Bohemian Forest) (Schönswetter et al., 2005; Albach, Schönswetter & Tribsch, 2006; Ronikier, Cieślak & Korbecka, 2008; Ronikier et al., 2008; Alvarez et al., 2009). However, in contrast to the Alps, the phylogeographical history of plants in the mountain ranges of Central Europe remains poorly understood (Ronikier, 2011). In addition, research related to the widespread circum-boreal plant species in the inhabited populations in the Sudetes, Bohemian Forest and Harz mountains is also scarce (Alsos et al., 2005; Kramp et al., 2009; Wróblewska, 2013; Jermakowicz et al., 2015). With the exception of studies that deal with Saxifraga (Bauert et al., 1998; Vargas, 2001; Lienert et al., 2002; Oliver, Hollingsworth & Gornall, 2006; Winkler et al., 2012; Winkler et al., 2013) and Salix sp. (Mirski et al., 2017), no studies have assessed the plants that inhabit the lowland or mountain peat bog ecosystems. Many peat bog plant species are present in a wide range of circum-boreal alpine regions and have been able to colonise large mountain areas or to migrate to North America via Beringia.

Populations that are dispersed in scattered localities are susceptible to negative effects that range from their individual population history to possible fluctuations in their size (number of individuals). Genetic drift and founder or bottleneck effects may also influence the genetic structure of populations that are often genetically isolated (Freeland, 2008; Hansen, Thomas & Arnholdt-Schmitt, 2009). The results of climatic oscillations has resulted in plant distributions and has also contributed to genetic diversity in populations that can be detected using molecular approaches (Alsos et al., 2005; Wasowicz et al., 2016). These methods are based on nuclear, ribosomal or chloroplast DNA (Taberlet et al., 1991) or DNA polymorphisms (Ronikier, 2011). These methods also provide detailed insight into the processes that are responsible for the distributions that are currently being observed and to describe the refugia areas for selected species. They also enable the detection and enhanced description of the genetic relationships between disjunctive and closed plant populations. This can help to define the microevolutionary processes that have occurred within the plant populations that have changed their previous distribution range or to identify divergent lineages (Ronikier, 2011).

Therefore the present study used DNA sequence data from chloroplast markers to investigate the genetic structure of individuals representing populations of the peat bog plant species, Swertia perennis, in the Polish lowlands, Carpathians, Sudetes, Bohemian Forest and Alps. The combination of samples from various localities together with cpDNA markers should permit the identification of different haplotype lineages across the 1,300 km disjunct distribution range in Europe. This study also aimed to assess whether (like in other temperate mountain species) e.g., from Central Asia) may be a valid explanation for the present distribution of S. perennis, and compared the genetic relationships between plant populations growing in geographically-isolated sites in natural or semi-natural patches in the landscape (Oostermeijer, 1996). These populations are commonly spatially isolated and thus likely differ genetically from each other (Young, Boyle & Brown, 1996). Therefore all of the studied S. perennis individuals were from populations that had a relatively wide but discontinuous distribution and were treated as isolated populations i accordance with the work of Lienert & Fischer (2004). We decided to test two hypotheses concerning migration of S. perennis: has migration occurred from ice-free areas of the Alps, or alternatievly plant migration occured from Eastern Europe or Asia. As a result of the tested hypothesis, closer phylogenetic relationships between the studied geographical regions should be visible in phylogenetical analysis.

Material and Methods

The species

S. perennis Linnaeus, Sp. Pl. 226, 1753, syn.: Gentiana palustris All. Fl. Pedem. 1: 100, 1785 from the Gentianaceae family inhabits lowlands but can also be found at higher altitudes. The species is distributed globally across the Northern hemisphere, but also shows discontinuous distribution from Asia to North America. It inhabits wetlands, especially peat bogs in calcareous fens or high mountains; however, it is also present in wet meadows, creek shores or wet rocks. It is a species of circum-boreal distribution, inhabiting Arctic, Siberian and Northern, Atlantic and Central European provinces (Hultén, 1968; Hultén & Fries, 1986; Meusel et al., 1978). In Europe, the species can be found in all mountain range systems of Alpine orogeny, including the Pyrenees, Caucasus, Dinaric Alps, Carpathians and Herzynian mountains (Sudetes, Bohemian Forest), as well as in peat bogs in Central Europe and Eastern lowlands. S. perennis is a long-lived perennial rhizome herb that usually produces one erect stem that grows to around 10–50 cm tall. It is a diploid (2n = 28) organism that flourishes in July or August (Löve & Löve, 1986; Pawlikowski & Wołkowycki, 2010). The flower nectar chambers are visited by various insects including species of Coleoptera, Lepidoptera, Diptera (especially Syrphidae) and Hymenoptera (especially Bombus and Vespidae). The plants develop up to 50 winged seeds in ovate fruit capsules (Lienert & Fischer, 2004). The species sometimes forms large populations in favourable habitats; however, as with numerous other peat bog plants, they are, but is also sensitive to habitat fragmentation and destruction, In the last century, fens in Poland and Switzerland that were inhabited by various wetland specialists (e.g., S. perennis) were strongly reduced in number and size due to direct destruction of wetlands, drainage and fertilisation (Lienert et al., 2002; Pawlikowski & Wołkowycki, 2010).

Study area and sampling

One plant sample per S. perennis population was collected from the Carpathians, Alps, Sudetes, Black Forest, Bohemian Forest and Polish lowlands close to the Lithuanian and Belarusian border (Fig. 1). In total, 10 plant samples were collected from the Carpathians, eight from the Sudetes, three from the Alps, two from the Bohemian Forest, one from the Black Forest and four from the Polish lowlands (Table 1). Fresh plant leaves were collected in the field, placed into plastic bags and immediately preserved using a drying agent (silica orange gel).

Figure 1 Sampling locations for the 28 populations of S. perennis examined in this study.

Geographical range of the species (after Meusel et al., 1978, changed) in Europe is indicated—a continuous line is marked compact geographical range; a dotted line is marked distributed range with isolated localities. The numbers are adequate to number of population from Table 1. Circles with different colors correspond to lineage divergences identified by TCS and phylogenetic analyze.

Table 1 Collection details of the studied plant material (S. perennis) used in the study.

No	Locality/Code	Region/Country	Altitude (m a.s.l.)	Latitude (N)	Longitude (E)	
1	Tengen, Schwarzwald	Schwarzwald/Germany	700	47°50′41.3″	08°39′08.7″	
2	Waldele, Allgäuer Alpen	Alps/Austria	1,116	47°21′27.3″	10°10′07.6″	
3	Schwende, Allgäuer Alpen	Alps/Austria	1,028	47°21′50.2″	10°10′37.5″	
4	Klausen Wald, Allgäuer Alpen	Alps/Austria	1,112	47°22′44.7″	10°11′22.8″	
5	Modravský potok, Šumava	Bohemian Forest/Czech Republic	1,121	48°58′11.2″	13°29′11.0″	
6	Luzenský potok, Šumava	Bohemian Forest/Czech Republic	1,143	48°57′18.2″	13°29′16.0″	
7	Sokolnik, Karkonosze	Sudetes/Poland	1,390	50°46′43.9″	15°31′36.2″	
8	Mały Śnieżny Kocioł, Karkonosze	Sudetes/Poland	1,380	50°46′59.5″	15°33′18.5″	
9	Kocioł Wielkiego Stawu, Karkonosze	Sudetes/Poland	1,295	50°45′43.5″	15°41′22.0″	
10	Kocioł Małego Stawu, Karkonosze	Sudetes/Poland	1,316	50°44′40.7″	15°41′57.6″	
11	Złote Źródło, Karkonosze	Sudetes/Poland	1,414	50°44′31.2″	15°42′57.2″	
12	Kopa, Karkonosze	Sudetes/Poland	1,276	50°44′41.1″	15°43′50.9″	
13	Kocioł Łomniczki, Karkonosze	Sudetes/Poland	1,234	50°44′23.9″	15°43′58.6″	
14	Slatinný potok, Hrubý Jeseník	Sudetes/Czech Republic	730	49°58′15.7″	17°12′03.1″	
15	Hala Cebulowa, Beskid Żywiecki	Carpathians/Poland	1,298	49°32′19.3″	19°18′48.2″	
16	Hala Miziowa, Beskid Żywiecki	Carpathians/Poland	1,282	49°32′24.7″	19°18′57.0″	
17	Dolné diery, Malá Fatra	Carpathians/Slovakia	660	49°15′01.5″	19°04′23.8″	
18	Veľký Rozsutec, Malá Fatra	Carpathians/Slovakia	1,618	49°13′53.5″	19°05′58.3″	
19	Dedošová dolina, Veľká Fatra	Carpathians/Slovakia	717	48°55′20.2″	19°02′13.7″	
20	Demänovská dolina, Nízke Tatry	Carpathians/Slovakia	950	48°59′50.6″	19°34′34.8″	
21	Roháčske plesá , Západné Tatry Tatry	Carpathians/Slovakia	1,646	49°12′31.6″	19°44′16.8″	
22	Dolina Kościeliska, Tatry Zachodnie	Carpathians/Poland	1,010	49°14′24.7″	19°51′50.2″	
23	Dolina Jaworzynka, Tatry Zachodnie	Carpathians/Poland	1,323	49°15′15.0″	19°59′51.1″	
24	Ostrva, Vysoké Tatry	Carpathians/Slovakia	1,647	49°09′05.2″	20°05′08.3″	
25	Kamień, Wyżyna Lubelska	Polish Lowland/Poland	181	51°06′28.3″	23°34′30.1″	
26	Biebrza, Kotlina Biebrzańska	Polish Lowland/Poland	126	53°38′34.5″	22°35′15.9″	
27	Kamienna Nowa, Kotlina Biebrzańska	Polish Lowland/Poland	131	53°42′35.2″	23°13′3.0″	
28	Rowele, Pojezierze Suwalskie	Polish Lowland/Poland	182	54°20′32.6″	22°54′56.6″	

DNA isolation and sequencing

The genomic DNA was isolated using the DNeasy Plant Mini Kit (Qiagen; Hilden Germany), according to the manufacturer’s protocol. Dried plant leaves were previously frozen using liquid nitrogen and disrupted from using Mixer Mill MM400 (Retsch; Haan, Germany). The quality and quantity of the DNA was determined using 1% TBE agarose gel.

In similar as Groff, Hale & Whitlock (2015), we sequenced three markers from the chloroplast genome of S. perennis: the trn L-trnF Intergenic Spacer, trnL Intron and the trnH(GUG) –psbA spacer. All three chloroplast DNA regions are widely used for phylogenetic studies at all taxonomic levels (Drábková et al., 2004). The trnLF region is often considered evolutionary conservative but employed in phylogeny and taxonomy, and some studies have found intraspecific variation in this biogeographically informative gene regions that has been contradicted (Taberlet et al., 1991; Brunsfeld & Sullivan, 2005; Shaw et al., 2005; Fujii & Senni, 2006; Shaw et al., 2007; Groff, Hale & Whitlock, 2015). Additionally, the trnH-psbA region of the cpDNA is often more variable than the trn LF region (e.g., Shaw et al., 2005). The trnL-trnF Intergenic Spacer together with trnL Intron were tested: with “c” and “f” primers (Taberlet et al., 1991) and trnH(GUG)–psbA with trnH(GUG) and psbA primers (Shaw et al., 2005). The DNA extracts were used to PCR andseqencing reactions. PCR reaction mix included (in the total volume of 20 µl): 1U Taq recombinant polymerase (Thermo-Fisher Scientific, Waltham, MA, USA), 10X Taq Buffer, one mM MgCl2, 0.5 µM of each primer, 0.4 mM dNTP and one µl DNA template. PCR cycle was performed with a Veriti Thermal Cycler (Life Technologies, Carlsbad, CA, USA) with the following parameters: 8 min at 95 °C, followed by 30 cycles of 45 s at 95 °C, 45 s at annealing temperature (49.2 °C –trnL, 51.2 °C –psbA) and 1 min at 72 °C, followed by a final extension step of 10 min at 72 °C. Prior to sequencing, PCR products were purified using GeneMATRIX PCR/ DNA Clean Up Purification Kit (Eurx, Gdańsk, Poland). Sequencing, post-reaction purification and reading were done by Genomed (Warsaw, Poland) using an ABI 377XL Automated DNA Sequencer (Applied Biosystems, Carlsbad, CA, USA). All sequences are available in GenBank (accession numbers - trnH(GUG) –psbA spacer: KY798346 –KY798347, KY817321 –KY817346; trnL-trnF Intergenic Spacer, trnL Intron: KY798346, KY798347, KY798348, KY906142 –KY906166). All molecular analyses was done at Department of Botany and Plant Ecology Wrocław University of Environmental and Life Sciences.

cpDNA data analyses

The cpDNA sequences were aligned using DNA Baser Sequence Assembler v4 (Heracle BioSoft, 2014) and checked for nucleotide variation using BioEdit ver. 7.1.11 (Hall, 1999). Combined both cpDNA region, a widely used for phylogenetic analysis at all taxonomic levels, were concatenated and analysed together. Prior to the phylogenetic analyses, the cpDNA sequences were aligned using Muscle software (Edgar, 2004a; Edgar, 2004b). We performed maximum parsimony (MP) and Bayesian inference (BI), to infer the phylogenetic relationships among selected individuals from European S. perennis populations. Maximum parsimony were conducted using PAUP* 4.0b10 (Swofford, 2002) and involved heuristic strategy with 1,000 replicates of random addition of sequences. Bootstraps for MP analyses based on 1,000 replications of full heuristic searches with the tree-bisection- reconstruction (TBR) branch-swapping algorithm, and those for NJ analyses (Saito & Nei, 1987) under the JC model (Jukes & Cantor, 1969). The BI analyses were performed using MrBayes 3.1.2. (Ronquist & Huelsenbeck, 2003). The substitution models used for each codon position in the BI analyses were GTR+G (first codon position), GTR+I (second codon position), and GTR+I+G (third codon position), as estimated based on Aikake’s information Criterion (AIC), selected by MrModeltest 2.3 (Nylander et al., 2004) using PAUP* 4.0b10 (Swofford, 2002). The parameters of the substitution models for codon position were unlinked. The Markov chain Monte Carlo iteration was performed and stopped at 1,000,000 generations. The first 25% of generations were discarded as burn-in, whereas the remaining trees were used to calculate a 50% majority-rule tree and to determine the posterior probabilities of individual branches. The remaining trees were used to produce a majority-rule consensus tree and to calculate posterior probabilities (PP).

Haplotype network of the studied cpDNA sequences were constructed by TCS v1.21:2 (Clement, Posada & Crandall, 2000). Results were also analyzed by DnaSP (Rozas et al., 2003): the haplotype diversity (Hd) and nucleotide diversity (π). Arlequin 3.5.1.2 (Excoffier & Lischer, 2010) was used for detection of genetic variation among groups, among plants representing populations within groups and molecular variation of haplotype distribution (Fst) (Weir & Cockerham, 1984).

In total, 28 sequences of S. perennis individuals and the sequences of five other species were used as an outgroup: Frassera speciosa, Lomatogonium rotatum, Comastoma tenellum and Gentianella amarella were studied.

Results

Molecular phylogenetic analyses using cpDNA sequence data

Alignment of the trnLF region of cpDNA revealed a variation in length among individuals from European populations. A total of 15 variable sites with insertions or deletions were identified. These sequences were 827–838 base pairs in length, of which 33 characters were parsimony informative (Table 2). Alignment of the trnH(GUG)–psbA cpDNA locus also revealed a variation in length among individuals as well as the studied sites. A total of 19 sites with insertions or deletions (Table 3) were identified. The sequences were 410–428 base pairs in length, of which 38 characters were parsimony informative. Several indels at 658–665 in the trnLF region and 70–87 in the trnH–psbA region were neglected because they varied inconsistently with the substitution and because indels typically mutate more frequently than substitutions (Alsos et al., 2005).

Table 2 Variable positions in the cpDNA trnLF region within S. perennis individuals.

Nucleotide position refer to the number of nucleotides from the first position of the region. The number and locality names correspond to Table 1.

No	Locality/Code	Length of trnLF region	12	24	25	37	38	61	71	147	195	241	556	650	775	784	796	
1	Tengen	831	–	C	–	A	C	T	C	–	T	–	C	G	–	–	G	
2	Waldele	830	T	C	–	A	–	T	C	–	T	–	T	G	–	–	A	
3	Schwende	832	T	C	–	A	C	T	C	–	T	–	T	G	–	–	A	
4	Klausen Wald	831	–	C	–	A	C	T	C	–	T	–	T	G	–	–	A	
5	Modravský potok	829	–	C	–	–	–	T	C	–	T	–	C	G	–	–	G	
6	Luzenský potok	827	–	C	–	A	–	–	–	–	T	–	C	G	–	–	G	
7	Sokolnik	831	–	C	–	A	C	T	C	–	G	–	C	G	–	–	G	
8	Mały Śnieżny Kocioł	830	–	C	–	C	–	T	C	–	T	–	C	G	–	–	G	
9	Kocioł Wielkiego Stawu	830	–	C	–	C	–	T	C	–	T	–	C	G	–	–	G	
10	Kocioł Małego Stawu	833	–	C	–	A	C	T	C	–	T	–	C	G	C	A	G	
11	Złote Źródło	831	–	C	–	A	C	T	C	–	T	–	C	G	–	–	G	
12	Kopa	832	–	C	T	A	C	T	C	–	T	–	C	G	–	–	G	
13	Kocioł Łomniczki	832	–	C	–	A	C	T	C	A	T	–	C	G	–	–	G	
14	Slatinný potok	830	–	C	–	C	–	T	C	–	G	–	C	G	–	–	G	
15	Hala Cebulowa	831	–	C	–	A	C	T	C	–	T	–	C	A	–	–	G	
16	Hala Miziowa	831	–	C	–	A	C	T	C	–	T	–	T	G	–	–	A	
17	Dolné diery	831	–	C	–	A	C	T	C	–	T	–	C	G	–	–	G	
18	Veľký Rozsutec	831	–	C	–	A	C	T	C	–	T	–	C	G	–	–	G	
19	Dedošová dolina	830	–	C	–	A	–	T	C	–	T	–	C	G	–	–	G	
20	Demänovská dolina	831	–	C	–	A	C	T	C	–	T	–	C	G	–	–	G	
21	Roháčske plesá	830	–	C	–	C	–	T	C	–	T	–	C	G	–	–	G	
22	Dolina Kościeliska	832	T	C	–	C	–	T	C	–	T	A	C	G	–	–	G	
23	Dolina Jaworzynka	830	–	T	–	C	–	T	C	–	T	–	C	G	–	–	G	
24	Ostrva	838	–	C	–	A	C	T	C	–	T	–	C	G	–	–	A	
25	Kamień	831	–	C	–	A	C	T	C	–	T	–	T	G	–	–	A	
26	Biebrza	831	–	C	–	A	C	T	C	–	T	–	C	G	–	–	G	
27	Kamienna Nowa	831	–	C	–	A	C	T	C	–	T	–	T	G	–	–	A	
28	Rowele	831	–	C	–	A	C	T	C	–	T	–	C	G	–	–	G	

Table 3 Variable positions in the cpDNA trnH-psbA region within S. perennis individuals.

Nucleotide position refer to the number of nucleotides from the first position of the region. The number and locality names correspond to Table 1.

No	Locality/Code	Length of trnH-psbA/ locus	1	4	15	19	55	88	97	99	124	126	127	143	144	145	246	277	286	399	411	
1	Tengen	429	A	G	C	C	–	–	C	–	A	–	T	T	A	A	A	C	T	T	A	
2	Waldele	429	A	G	C	C	–	T	A	–	A	–	T	G	T	T	A	A	T	G	A	
3	Schwende	411	A	G	C	C	–	T	A	–	A	–	T	G	T	T	A	A	T	G	A	
4	Klausen Wald	429	A	G	C	C	–	T	A	–	A	–	T	G	T	T	A	A	T	G	A	
5	Modravský potok	411	A	G	C	C	–	–	C	–	A	–	T	T	A	A	A	C	T	T	A	
6	Luzenský potok	412	C	G	C	C	–	–	C	–	A	T	T	T	A	A	A	C	T	T	A	
7	Sokolnik	410	A	G	C	C	–	–	C	–	A	–	–	T	A	A	A	C	T	G	A	
8	Mały Śnieżny Kocioł	413	A	G	C	C	–	–	C	T	A	T	T	T	A	A	A	C	T	T	A	
9	Kocioł Wielkiego Stawu	412	A	T	A	G	–	–	C	–	A	T	T	T	A	A	A	C	T	T	A	
10	Kocioł Małego Stawu	412	A	G	C	C	–	–	C	–	A	T	T	T	A	A	A	C	T	T	A	
11	Złote Źródło	410	A	G	C	C	–	–	C	–	A	–	–	T	A	A	A	C	A	G	A	
12	Kopa	410	A	G	C	C	–	–	C	–	A	–	–	T	A	A	A	C	A	G	A	
13	Kocioł Łomniczki	412	A	G	C	C	–	–	C	–	A	T	T	T	A	A	A	C	T	T	C	
14	Slatinný potok	410	A	G	C	C	–	–	C	–	A	–	–	T	A	A	A	C	T	G	A	
15	Hala Cebulowa	412	A	G	C	C	–	–	C	–	A	T	T	T	A	A	A	C	T	T	A	
16	Hala Miziowa	428	A	G	C	C	–	T	A	–	A	–	–	G	T	T	A	A	T	G	A	
17	Dolné diery	410	A	G	C	C	–	–	C	–	A	–	–	T	A	A	A	C	T	G	A	
18	Veľký Rozsutec	410	A	G	C	C	–	–	C	–	A	–	–	T	A	A	A	C	T	G	A	
19	Dedošová dolina	410	A	G	C	C	–	–	C	–	A	–	–	T	A	A	A	C	T	G	A	
20	Demänovská dolina	412	A	G	C	C	A	–	C	–	A	–	T	T	A	A	A	C	T	T	A	
21	Roháčske plesá	410	A	G	C	C	–	–	C	–	A	–	–	T	A	A	A	C	T	G	A	
22	Dolina Kościeliska	411	A	G	C	C	–	–	C	–	A	–	T	T	A	A	A	C	T	T	A	
23	Dolina Jaworzynka	411	A	G	C	C	–	–	C	–	A	–	T	T	A	A	A	C	T	T	A	
24	Ostrva	423	A	G	C	C	–	A	C	–	T	–	–	T	A	A	C	A	T	G	A	
25	Kamień	428	A	G	C	C	–	T	A	–	A	–	–	G	T	T	A	A	T	G	A	
26	Biebrza	410	A	G	C	C	–	–	C	–	A	–	–	T	A	A	A	C	T	G	A	
27	Kamienna Nowa	428	A	G	C	C	–	T	A	–	A	–	–	G	T	T	A	A	T	G	A	
28	Rowele	410	A	G	C	C	–	–	C	–	A	–	–	T	A	A	A	C	T	G	A	

The aligned lengths of the 33 sequences of the trnLF and trnH–psbA regions varied from 1,485 to 1,503 base pairs. The whole dataset contained 36 variable sites that were dispersed randomly across the entire analysis area. The results showed that maximum parsimony (MP) analyses were congruent with Bayesian interference (BI) analyses. The concatenate sequence data were more informative than single trees based on the trnLF and trnH–psbA regions, and the results are presented in Fig. 2. The topologies of the trees were congruent, and only one arbitrarily-selected tree (from 12 trees) is shown with bootstrap proportions (BP) from MP and BI at the nodes.

Figure 2 Phylogenetic relationships among haplotypes and lineages detected in S. perennis.

The phylogenetic tree is based on studied trnL-trnF and trnH-psbA cpDNA sequences. Bootstrap values of MP and BI analysis are given close to branches, respectively.

Phylogeographic analysis of the concatenate sequence data resulted in more than 200 parsimonious trees (CI = 0.94, RI = 0.92). The dataset did not reveal any distinct regions of S. perennis within Europe, and no congruent groups were identified using TCS software (Fig. 3). Several individuals from distinct populations had unique sequences; however, there was no clear connection with geographical structure. Several of the most common haplotypes were found in individuals in different populations from various regions (e.g., in individuals from the Polish lowlands, Carpathians and Sudetes) (Fig. 3). In clade (BS = 67, BP = 0.98) are placed individuals represented 11 populations from almost all study regions. To verify the results, numerous additional resequencing reactions were performed on the same or additional S. perennis samples.

Figure 3 Maximum Parsimony networks analysis of cpDNA haplotypes identified by TCS software.

Solid lines between circles represent one mutational step between two chlorotypes based on most parsimonious algorithm. The small open circles indicate the missing chlorotypes (not sampled or extinct). Circle colors correspond to haplotype lineages, respectively, as shown in Fig. 2.

When assessing genetic polymorphisms at the species level, S. perennis possessed high levels of plastid DNA diversity (HD = 0.841) and nucleotide diversity (π = 0.00154). Analysis of molecular variance (AMOVA) indicated a high degree of differentiation among individuals from populations (Fst = 0.649) and a high level of differentiation (64.93) among groups of populations (Table 4). Differentiation among individuals from populations within groups was lower and reached 35.07. This confirms that the determined haplotypes are dispersed randomly across the whole of Europe.

Table 4 Results of analysis of molecular variance of S. perennis examined in this study.

Source of variation	d.f.	Sum of squares	Variance component	Percentage of variation	
Among groups	5	22.405	0.87418	64.93	
Among populations within groups	22	10.388	0.47217	35.07	
Total	27	32.792	1.34636		
Notes.

Fst = 0.649.

Discussion

Genetic variability in plant populations is influenced by complex historical processes (e.g., Pleistocene glaciations, migrations, bottleneck effects and gene flow) and biological processes, the results of which can be observed using genetic analyses. However, it is difficult to define the same rules for all plant species, particularly with respect to plant migration. A good example is the history of S. perennis in Central and Western Europe, as this species is believed to have migrated there from Central Asia as another species from the Gentianaceae family. This study was not able to confirm this hypothesis using the data acquired. In fact, the findings suggest a discontinuous expansion range from various directions (e.g., Central Asia and Northern Euro–Asian territories). The findings also illustrate cpDNA variation between individuals from different S. perennis populations in Europe, with numerous unique haplotypes dispersed across this geographical area. A clear phylogeographic structure of S. perennis was not detected in Europe, and the results illustrate distinct structural differences between geographical haplotype lineages and the complicated nature of preglacial and postglacial plant dispersal (Abbott & Brochmann, 2003). The observed haplotypes were dispersed randomly across the study area and did not form closely-related lineages among geographical regions (Figs. 2 and 3). These results show that the S. perennis haplotypes probably originated from different localities and not only from one direction (e.g., Central Asia). They could also have arisen before they spread to their current localities, from where they dispersed and mutated. Plants collected from Hala Cebulowa and Hala Miziowa were situated approximately 500 m apart, and their haplotypes differed with regards to trnH–psbA length and nucleotide variation in the trnH–psbA and trnLF regions. Similarly, the haplotypes detected in plants from Karkonosze populations (Sudetes mountains) were located about one km from each other (Kocioł Wielkiego Stawu–Kocioł Małego Stawu and Kocioł Łomniczki–Kopa) and they also differed in terms of their trnH–psbA and trnLF length or nucleotide variation. Other haplotypes were specific only for the sites or localities in which they were grown and did not form groups of similar sequences (Fig. 2). Closely-located populations of S. perennis appear to be genetically isolated and form isolated population systems. This is in accordance with the work of Lienert et al. (2002) who suggested that geographical and genetic distances between S. perennis populations were not correlated and that gene flow between close populations was not higher than between more distant populations.

The genetic diversity of cpDNA and the lack of phylogeographical structure may be attributed to the range formation history, its shifting during the ice age and the emergence of a group of boreal mountain plants (Ronikier, 2011). During subsequent glaciations and cold temperatures, alpine plants expanded their range and warm interglacial alpine plants and boreal and subarctic species (from subpolar areas in front of the glacier) retreated to higher mountain localities and refugia areas in the far north. This may have led to distinctive geographical disjunctions. Today, S. perennis is found in European mountains, Northern

Eurasia and North America (in peat bogs and boreal forests), as well as in isolated boreal (Hultén, 1968; Meusel et al., 1978; Hultén & Fries, 1986).

Similar haplotypes identified in this study may also form part of a residual lineage of haplotypes that colonised peat bog areas in mountains. The five most similar haplotypes (17, 18, 19, 26 and 28, presented on Fig. 3) are dispersed across several disjunctive areas without any geographical correlation. The current fragmented distribution of S. perennis seems to be a residual effect after more homogenous distribution. It is also possible that cpDNA mutates rapidly, seed production or dispersal does not occur or is scarce and gene flow is scattered or nonexistant. At the contact zones between migrating fronts, significant haplotype mixing has occurred, as all populations are colonising new territories. It is possible that the migration of S. perennis occurred in a similar way or in several phases. Mountain plants could have survived the last Wistulian glaciation; however, after the end of the cold period, new plants (possibly from Siberia) may have expanded to their present European territory. This may explain the difference in cpDNA nucleotide composition in individuals from distinct populations of S. perennis. It is also possible that the species both survived glaciations in situ and remigrated several times, therefore broad-fronted repeated recolonization and glacial survival can shape genetic variation (Tausch et al., 2017).

Conclusions

In conclusion, the non-existent phylogeographical structure of S. perennis in the study areas of the European range may be the result of numerous overlapping factors, such as multidirectional gene flow in the dispersal history, long-distance dispersal during postglacial recolonization and survival in several detached refugia (Beatty & Provan, 2011; Cain, Milligan & Strand, 2000; Jiménez-Mejías et al., 2012; Sanz et al., 2014). The low level of variation in the structure patterns is similar to that of other plant taxa with northern distribution and may indicate the long-term process of gene flow among the populations (Eidesen et al., 2007a; Eidesen et al., 2007b; Ehrich, Alsos & Brochmann, 2008; Westergaard et al., 2010; Alsos et al., 2012). It is also possible that plant populations occurred in Pleistocene refugia or migrated during the Holocene, and this was enough time for them to form divergent cpDNA. The lack of evidence for phylogeographic structure may indicate a high level of gene flow in the recent past. The variation in cpDNA nucleotide composition in individuals from distinct populations may also reflect genetic variability from ancient periods when the landscape and fen systems were not fragmented. The gene flow today is probably much smaller than in previous times or is non-existent. Habitat fragmentation for peat bog specialists may have significantly reduced genetic variability and led to absent or minimal gene flow between populations.

Supplemental Information

Supplemental Information 1 Rough DNA sequence data used in the analysis

Click here for additional data file.

We would like to thank to Maria Kwiatkowska for her assistance in the laboratory and the two anonymous reviewers for their comments which helped to improve previous versions of this manuscript.

Additional Information and Declarations

Competing Interests

Author Contributions

Data Availability

The authors declare there are no competing interests.

Jacek Urbaniak conceived and designed the experiments, performed the experiments, analyzed the data, contributed reagents/materials/analysis tools, prepared figures and/or tables, authored or reviewed drafts of the paper, approved the final draft, sample collection.

Paweł Kwiatkowski prepared figures and/or tables, authored or reviewed drafts of the paper, approved the final draft, sample collection.

Paweł Pawlikowski authored or reviewed drafts of the paper, approved the final draft, sample collection.

The following information was supplied regarding data availability:

All sequences are available in GenBank (accession numbers: trnH(GUG) – psbA spacer: KY798346–KY798347, KY817321–KY817346; trnL-trnF Intergenic Spacer, trnL Intron: KY798346, KY798347, KY798348, KY906142–KY906166).

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
