# Peer review of "Phylogeography of Swertia perennis in Europe based on cpDNA markers"

_PeerJ, doi:10.7717/peerj.5512_

## Round 0.1 · original submission · Major Revisions

Both reviewers agree that your paper includes data that warrant publication but there are several substantive points that need to be addressed in a revision. Both believe that the study is compromised by a low sample size (within and across populations) and both recommend that the study is more clearly framed as a research question or hypothesis. I agree with them. Please address these and other concerns raised by the reviewers in your revision.

Reviewer 1 ·

Basic reporting

The documentation and presentation is very good, most parts are well written, but there are many language errors (article use, singular/plural) and also typos.

Otherwise documentation and main parts are well written although there are some language issues with the grammar which would need attention

There are plenty of language problems – e.g.:
Line 55: level of gene flow in the recent past?, line 58: difficult to speculate about relations?? 118 Taberlet, 137: inhabits, 148 flourishEs.350 formation of divergent cpDNA? 355 specialist sp.?

Experimental design

The authors provide a phylogeographical study based on cpDNA variation of populations of Swertia perennis, a species of minerotrophic bogs and alpine wet grasslands widespread in Eurasia, but disjunctly distributed in Europe. The authors studied mainly populations of European mountains although the species is widely distributed in Eurasia. Despite haplotypic variation found the authors could not confirm any clear genetic and phylogeographical structure, which does not allow strong conclusions.

Validity of the findings

The authors evidenced that there is a lack of phylogeographical patterns and raise several possibilities (postglacial immigration, many refugia,…) as explanation. As the sampling is uneven I would recommend to provide the total distribution of the species in the study area. This should give additional support for the “lacking pattern”.
I can follow more or less that there is a (complete?) lack of phylogeographic structure, nevertheless they could test hypotheses. It might me worthwhile to test the hypothesis that (like in other alpine species) late/postglacial immigration from Central Asia might be a valid explanation for the pattern observed. Swertia perennis is a very common species in large parts of Eurasia. I am also not fully convinced that there is no phylogeographical structure, the three Alpine samples cluster in one clade.
I suggest that the authors provide at least summary statistics (e.g., using Arlequin) to show lack of pattern (e.g. using AMOVAS of regions). Also other statistical phylogeographical approaches could be taken into account. (testing for expansion, etc.).

The authors state that “clonality” could have an effect, were more than sample per population analyzed? This is unclear. I understood that only one sample per population was sequenced, this no conclusions on clonality could be drawn.

Reviewer 2 ·

Basic reporting

The presented manuscript of Urbaniak et al. addresses phylogeographic differentiation of Swertia perennis, the plant species occurred in peat bogs of northern hemisphere. The authors based their study on the cpDNA markers to elucidate the history of its nowadays distribution. As the information of the phylogeography of wetland species in Europe is still insufficient, the paper brought some news to our knowledge of the genetic differentiation of this group of species. The methods used seems to be adequate and the manuscript seems to be well structured. However, the manuscript needs to be improved before an acceptance to the publishing. The authors used only one individual per population and they should be more aware of their hypotheses thorough the text. Every time they should use something like “the individuals represented populations” .. not only “populations”. Additionally, they discuss the genetic position and potential history of lowland populations but they used only four populations - also here they should be more cautious. One part of the hypothesis is connected with isolated site or isolated population but we have no definition of these terms and no list of the individuals (populations) linked with these terms.
In some parts of the text the authors are not congruent. There is not clear what authors treat as one haplotype – sometimes they used parsimony network, sometimes phylogenetic tree. It could be explain and consistent. Maybe they should list somewhere in the results the numbers of localities shared the same haplotypes or put it in the Table 1.
The manuscript should be also linguistically and grammatically improved. As the English is not my mother language I do not feel good enough to improve it.
The authors should check carefully the German, Slovak, Czech last names and names of localities for diacritical marks – as there are many differences between references and the main text; in the table 1 there is any for Slovak localities.

Experimental design

Original primary research within Aims and Scope of the journal.

The submission should clearly define the research question, which must be relevant and meaningful - see the comment in the basic reporting section.

Rigorous investigation performed to a high technical & ethical standard.
Methods should be described with sufficient information to be reproducible by another investigator - see the comment in the basic reporting section.

Validity of the findings

Conclusion are well stated, linked to original research question & limited to supporting results, but see in the basic reporting.
Speculation is welcome, but should be identified as such, .

Additional comments

For the authors:
The presented manuscript of Urbaniak et al. addresses phylogeographic differentiation of Swertia perennis, the plant species occurred in peat bogs of northern hemisphere. The authors based their study on the cpDNA markers to elucidate the history of its nowadays distribution. As the information of the phylogeography of wetland species in Europe is still insufficient, the paper brought some news to our knowledge of the genetic differentiation of this group of species. The methods used seems to be adequate and the manuscript seems to be well structured. However, the manuscript needs to be improved before an acceptance to the publishing. The authors used only one individual per population and they should be more aware of their hypotheses thorough the text. Every time they should use something like “the individuals represented populations” .. not only “populations”. Additionally, they discuss the genetic position and potential history of lowland populations but they used only four populations - also here they should be more cautious. One part of the hypothesis is connected with isolated site or isolated population but we have no definition of these terms and no list of the individuals (populations) linked with these terms.
In some parts of the text the authors are not congruent. There is not clear what authors treat as one haplotype – sometimes they used parsimony network, sometimes phylogenetic tree. It could be explain and consistent. Maybe they should list somewhere in the results the numbers of localities shared the same haplotypes or put it in the Table 1.
The manuscript should be also linguistically and grammatically improved. As the English is not my mother language I do not feel good enough to improve it.
The authors should check carefully the German, Slovak, Czech last names and names of localities for diacritical marks – as there are many differences between references and the main text; in the table 1 there is any for Slovak localities.

Some detailed suggestions:
L45: The cpDNA were analysed to detect genetic relationship between…
L47: 20 haplotypes were revealed across the dataset. They …..
L54-55: the sentence is not congruent with the text: see l.280-286, 301-303, 323– it should be deleted or written in another way.

Introduction:
L.63: ..are a consequence of..
l.66: Hewitt….; Taberlet et al....
l.67: ... in the front of…
l.69: Slovák ..
l.90: …potential refugia for …
l.90-91: …in fragmented subranges in several countries: - rewrite - it should be written regard to regions not countries.
l.106: Winkler et al., 2012;
l.105-107: the authors should mention here the article of Swertia by Lienert at al (2002) at some point.
l.111: The populations …
l.114: …populations which..
l.118: Taberlet …
l.125: .. for investigation the genetic..
l.131: this part of the hypothesis is not well constructed as the authors did not defined and did not listed what they considered as isolated population or isolated site for Swertia perennis in their study. They should explain it and define or change (delete?) this question.

The species
l.136-137 – somewhere should be done the family of the species.
l.137: …is a highly morphologically variable taxon, …
l.140: ..but it is also..
l.147: ..cm
l.148: It is a diploid…
l.148-149: The information about pollination system, fruits and dispersal mode should be done.
l.149-150: I will advice at least to cite Lienert, Fischer, Diemer (2002: Local extinctions.. ) and Pawlikowski & Wołkowycki (2010) according to the disappearance of the species localities (or put some sentences with citation).


Study area & sampling
l.155-160. The authors have to directly write that it was one individual per population. They should also write about their idea of sampling – as they write in l.128 – 1300 km disjunctive distribution, some individuals were sampled in closed populations (see l.294-295) etc
l.159: tissue=leaves?, ……in to=into
l.160: persevered?

DNA isolation….
l.173: ….in these …, ….biogeographically…
l.176: trnLF?
l.179: The DNA extracts were used to …..
l.184: …45 s min?
l.192: analyses

cpDNA data…
l.199: should be here explain why both cpDNA regions (see l.170: three chloroplast…)
l.220: this sentence should be rewrite to be more informative (e.g. In total, 28 sequence of S. perennis individuals and sequences of five other species used as an outgroup……)

Results:
The spaces between numbers – be consistent with the journal requirements
l.228: …populations.
l.236: …varied from?
l.245: dataset
l.246-247: rephrase the sentence
l.250: ..populations
l.250. …One of them represents a common haplotype … There is not exact as we see on the Fig. 3 two haplotypes: 16,25,27 and 3,4. I think that the authors should be more consistent with using “haplotype” in this paragraph and use word clade or another word for the clade of the tree (eg. L.251-253: Other clade combines two haplotypes: ….). Maybe the authors should rewrite this paragraph to be more clear for readers. I am not sure from the text and figures what authors treat as type of haplotype (what is the primary analysis?).
I will also advice to put the names/localities or numbers of some samples.
l.256: rewrite sentence: not populations but individual represented population.
l.256/7: most of them (how many?)
l.258: ..of the same and/or other …. – what does it mean? If the authors have more than one specimen per population the results will be more convinced. They also should put this information in the method section.
l. 259 ..across the studied part of the European range (or something like that)

Discussion
l.256: analyses
l.267-270 – I am not convinced if the example of R. pygmaeus is good here as the authors in the next sentence write of the origin of Swertia perennis from Central Asia. Next sentences (l.270) do not prove this hypothesis, rather more complicate it. Maybe the authors should use better explanation of their hypothesis (like diversity of genus Swertia or better explanation of distribution of S.perennis).
l.272: across its studied part of ….
l.276: .. based on cpDNA from populations existed in ….(geographical regions)
l.276-277: the authors did not studied polymorphism within populations! – this sentence should be rewritten or delete. Surely, they should refer to the paper of Groff et al but be aware of the dataset (geographical regions, number of samples per population)
l.277-280: delete – it is written l.258..
l.279: analyses
l.282-286: this part of the text should be clarified; it should be clearly written that this are the results of Groff et al. In first reading it seems that the authors studied polymorphism within populations (more than one individual per population) and they did not observe seedlings.
l.289: please change ..the whole of Europe ..
l.290: ..cpDNA of …
l.293: ..among close (neighbouring) geographically?
l.294: Hala Cebulowa … should be in the new sentence with region listed.
l.293-300: should be rewritten to clarify – it is very interesting that close localities are so differentiated
l.311: (any citation?)
l.318: (17,18,19,26,28)
l.319 – delete “the Alps” – none of the listed samples d not originate from the Alps.
l.320-322: the authors should be aware here of the glaciations history of the polish lowlands. Maybe the 25 could be the older then 26-28. Maybe the authors will find the palynological papers from these part of Poland with the history based on peat bogs. If so, it could bring a new insight.
l.324-326. I am not sure if the example of Quercus species is here good – there are several species and they are woods.
l. 326: Dumoulin-Lapègue
l.335-337: these sentences should have placement (detailed information of subspecies) in The species section or should be deleted.

Conclusion
l.341-355: some of the sentences should be rewritten according to changes in the main text.
l.341: in Europe: in studied part of European range
l.346: other northern taxon types?= other plant taxa with northern distribution?
l.347: populations
l.348: Alsos et al., 2012).
l.350: cpDNA
l.355 sp.??

References
The authors should carefully check the cited positions in accordance to: diacritical marks, letters of names, letters in titles (small/capital), dots after journal types, comma after last name, spaces.
Here are some examples that I noticed:
l.369, 372, 373, 376, 408, 430, 435, 465, 497, 502, 518, 565, 568, 571.

Figures
Fig. 1: Distribution of studied populations of Swertia perennis…. …..analyses.
(it would be nice to have distribution range shaded)
Check the color for 3, 4 – according to Fig. 3 it should be different and, of course, consistent through all figures.

Fig. 2 …Swertia….

Fig. 3. ….of Swertia perennis.
The authors should be consistent with the size of the circles: too large for 1,23, 3,4, 7. Be aware of the colors as it was mentions above.

Table 1. Sampling localities for the studied individuals of Swertia perennis across the European range of the species.
The authors should use the diacritical markers for the localities names. The columns 2, 3 should be left aligned.
27: Kotlina??

Table 2 and 3: Variable positions in the cpDNA t….region within Swertia perennis individuals. ….The number and locality names correspond to Table 1.

---

## Round 0.2 · Minor Revisions

Thank you for your revised manuscript. While the paper is technically sound, the language requires substantive improvement before it is acceptable for an English language journal. You will see on the attachment that I have identified several sentences in the Introduction which require improvements in language. I have also edited the Conclusions. I suggest that you persuade a colleague who is both proficient in English and in plant phylogeography to edit the entire manuscript. I look forward to seeing the revision.

---

## Round 0.3 · accepted · Accept

I am satisfied that the language has been sufficiently improved and other minor technical issues have been addressed.

#